# Exploiting evolutionary steering to induce collateral drug sensitivity in cancer

Ahmet Acar[1,9,10], Daniel Nichol [1,10], Javier Fernandez-Mateos[1], George D. Cresswell [1], Iros Barozzi[2], Sung Pil Hong[2], Nicholas Trahearn[1], Inmaculada Spiteri [1], Mark Stubbs[3], Rosemary Burke[3], Adam Stewart[4], Giulio Caravagna[1], Benjamin Werner [1], Georgios Vlachogiannis [5], Carlo C. Maley [6], Luca Magnani [2], Nicola Valeri[5,7], Udai Banerji [3,4,8✉] & Andrea Sottoriva [1✉]

Drug resistance mediated by clonal evolution is arguably the biggest problem in cancer therapy today. However, evolving resistance to one drug may come at a cost of decreased fecundity or increased sensitivity to another drug. These evolutionary trade-offs can be exploited using 'evolutionary steering' to control the tumour population and delay resistance. However, recapitulating cancer evolutionary dynamics experimentally remains challenging. Here, we present an approach for evolutionary steering based on a combination of single-cell barcoding, large populations of $10^8$–$10^9$ cells grown without re-plating, longitudinal non-destructive monitoring of cancer clones, and mathematical modelling of tumour evolution. We demonstrate evolutionary steering in a lung cancer model, showing that it shifts the clonal composition of the tumour in our favour, leading to collateral sensitivity and pro-liferative costs. Genomic profiling revealed some of the mechanisms that drive evolved sensitivity. This approach allows modelling evolutionary steering strategies that can poten-tially control treatment resistance.

[1] Evolutionary Genomics and Modelling Lab, Centre for Evolution and Cancer, The Institute of Cancer Research, London, UK. [2] Department of Surgery and Cancer, Imperial College London, London, UK. [3] CRUK Cancer Therapeutics Unit, The Institute of Cancer Research, London, UK. [4] Clinical Pharmacology—Adaptive Therapy Group, Division of Cancer Therapeutics and Clinical Studies, The Institute of Cancer Research, London, UK. [5] Gastrointestinal Cancer Biology and Genomics Team, Centre for Evolution and Cancer, The Institute of Cancer Research, London, UK. [6] Arizona Cancer Evolution Center, Biodesign Institute, Arizona State University, Tempe, USA. [7] Department of Medicine, The Royal Marsden NHS Foundation Trust, London, UK. [8] Drug Development Unit, The Institute of Cancer Research and The Royal Marsden Hospital NHS Foundation Trust, London, UK. [9] Present address: Department of Biological Sciences, Middle East Technical University, Ankara, Turkey. [10] These authors contributed equally: Ahmet Acar, Daniel Nichol. ✉email: udai.banerji@icr.ac.uk; andrea.sottoriva@icr.ac.uk

Although targeted cancer therapies are effective in many patients[1], complete eradication of the disease is impeded by treatment resistance, currently an intractable problem in cancer. Resistance is often mediated by redundancies in downstream signalling pathways[2], cell phenotypic plasticity[3], and most importantly, intra-tumour heterogeneity (ITH)[4]. High levels of ITH, and the huge number of cells in a tumour, imply that pre-existing cancer subclones that are drug resistant because of heritable genetic[5] or epigenetic[6,7] alterations are invariably present when treatment starts[8,9], thus leading to Darwinian adaptation[10]. Treatment resistance can also be polyclonal, with multiple distinct subclones harbouring different resistance mechanisms driving tumour progression, thus making resistance even harder to control[11]. In addition, drug-tolerant cancer cells or 'persistors' can survive and acquire de novo alterations that give rise to fully resistant subclones during or after treatment[12,13]. The emergence of pre-existing populations that prior to treatment are fitness neutral (or even deleterious)[14] and are positively selected by intervention can be recapitulated in the lab, as first demonstrated by the classic Luria–Delbrück experiment in bacteria[15].

Hence, cancers are unlikely to be successfully treated with a single agent[16]. Whereas combination strategies are often highly toxic and impractical, relatively little is known about the most effective sequence of drugs. Administering a drug can sensitise cancer cells to a second drug, a phenomenon known as collateral sensitivity, which has been demonstrated in seminal studies in bacteria[17–19], malaria[20] and cancer[14,21,22]. This is based on the observation that, as in ecological systems, developing a new trait, such as resistance to cancer treatment, likely comes at the expense of other features[23,24], leading to a trade-off often referred to as an 'evolutionary double bind'[25,26]. Indeed, cost of resistance has been observed in distinct pathogenic organisms[27] as well as in cancer[28,29].

Evolutionary steering refers to the use of drug intervention aimed at exploiting trade-offs to control tumour evolution. The goal is directing the evolution of the tumour population using Darwinian adaptation to a drug. When a second drug is administered, the clonal composition of the population is different from the start, and this can lead to increased sensitivity, or even complete extinction[30,31]. In this scenario, because steering has changed the clonal structure of the population, collateral drug sensitivity is likely to be persistent rather than transient. Therapeutic strategies that rely on rational evolutionary steering to control clonal evolution are likely less subject to stochastic temporary effects and cell plasticity, and hence potentially more feasible to implement in the clinic.

Here we present an approach for evolutionary steering based on a combination of single-cell barcoding, very large populations of $10^8$–$10^9$ cells grown without re-plating, longitudinal non-destructive monitoring of cancer clones, and mathematical modelling of tumour evolution. We use this method to quantitatively study evolutionary steering and demonstrate the evolutionary determinants of collateral drug sensitivity in cancer cell populations.

## Results

**Recapitulating the evolution of cancer drug resistance experimentally.** Standard experimental approaches are inadequate to study evolutionary steering because they are limited to small populations that do not recapitulate the extensive ITH present in human malignancies[32]. Current methods to study drug resistance rely on passaging the cells repeatedly under low drug Effective Concentration (e.g. EC50) or escalating doses until a fully resistant phenotype arises, 6 months to a year later[33]. Although these techniques are useful to generate fully resistant lines, the temporal evolutionary dynamics in these systems are unlike what happens in patients. First, current techniques are based on waiting for a de novo mutation, rather than selecting for a pre-existing subclone (Fig. 1a). This means that in each replicate the evolutionary process is different, driven by the stochastic arrival of a new mutation, which has very variable waiting times, as we demonstrate with stochastic simulations in Fig. 1b (see "Methods" section). It is not even guaranteed that the same resistance mutation arises in each replicate, thus making replicates hard to compare and results difficult to generalise. Second, re-plating induces sampling bottlenecks that are hard to control, leading to genetic drifting and artificial loss of ITH through time (Supplementary Fig. 1A). Mutation rates per base per cell division in cancer are in the order of $10^{-8}$–$10^{-7}$ (refs. [21,30,34]), and with standard 384-well plates or even T175 flasks, not even 10 passages (assuming 1:10 re-plating) are enough to have a single mutant in the population (Supplementary Fig. 1b), and instead 10 times the population of a T175 flask is required (see "Methods" section for details on the calculations). Hence, small culture systems risk pushing the evolutionary dynamics towards highly stochastic regimens that are very hard to predict and control. Moreover, cell plasticity and drug tolerance[35] are important mechanisms of drug adaptation, leading to resistance that is non-heritable and potentially reversible[6]. Non-heritable drug resistance can arise through epithelial–mesenchymal transition[27,36,37] or upregulation of drug-efflux pumps[38]. In small cultured populations driven by stochastic forces and de novo mutants, it is extremely hard to distinguish the heritable and non-heritable components of drug resistance.

Here, we present an experimental approach that leverages on large populations ($>10^8$) containing trackable pre-existing resistant subclones with highly reproducible evolutionary dynamics (Fig. 1c) to overcome the limitations of standard approaches (Fig. 1d).

**Evolutionary steering of resistant cells through fitness landscapes.** The relationship between heritable genetic or epigenetic information, and the corresponding cellular phenotype, can be represented by the classical fitness landscape model[39]. Phenotypes are multifaceted and arise as a product of the complex interactions between heritable factors and the environment. If we summarise the fitness of these complex phenotypes with respect to a certain condition or environment by a single value, the genotype–phenotype relationship can be represented as an $n + 1$ dimensional space whereby the alleles present at $n$ (epi)genetic loci are mapped to the relative fitness advantage they confer. A single cell can therefore be represented by a point in this landscape corresponding to its (epi)genetic state. As populations proliferate and randomly mutate, cell lineages move around the landscape. In a simple illustrative drug-free scenario (Fig. 1e), multiple cells, each characterised by a certain genotype ($x$, $y$ and $z$), are scattered around a neutral 'flat' fitness landscape because of mutations. When a drug is applied (e.g. drug 1), the fitness landscape changes, and genotypes that were previously neutral (or even slightly deleterious) may become advantageous under the new condition (e.g. $y$ and $z$), and outcompete the rest (e.g. $x$). Due to Darwinian selection, populations in lower fitness elevations will likely go extinct, whereas populations in fitness 'peaks' will prosper. This makes populations 'climb' higher and higher fitness peaks, leading to evolutionary adaptation.

Different drugs may select for distinct phenotypes (e.g. $y$ and $z$ are differentially selected by drugs 2 and 3—Fig. 1e). Using drugs with divergent fitness landscapes is the central idea of evolutionary steering. This concept is illustrated in Fig. 1f. Tumourigenesis gives rise to a heterogeneous population of

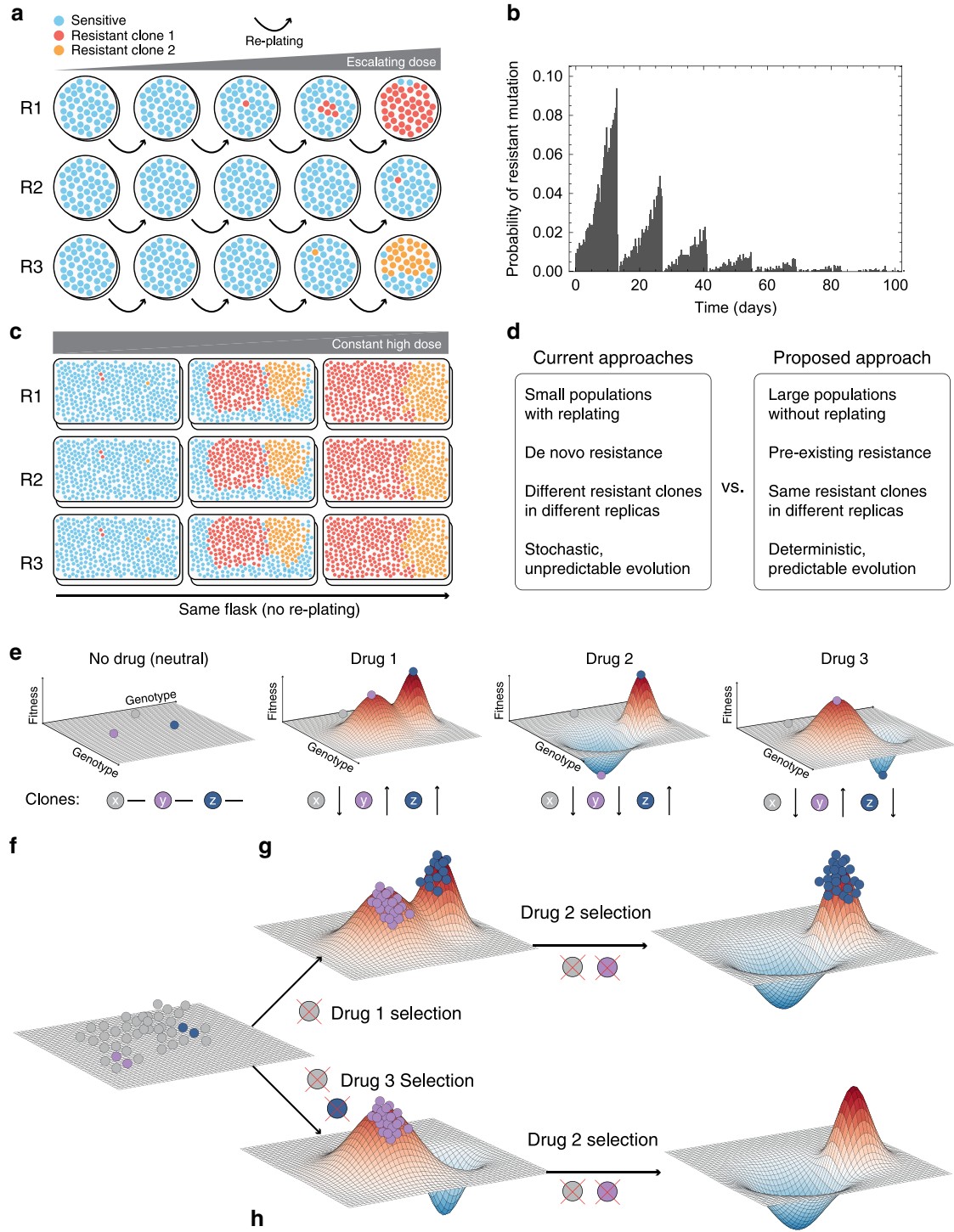

**Fig. 1 Limits of current approaches to study the evolution of cancer drug resistance through fitness landscapes. a** Current approaches use small populations with re-plating and escalating drug dose to produce de novo resistant lines over a period of 6–12 m. Evolutionary dynamics in this setting are highly stochastic, with different mutants arising at distinct times in different replicates. **b** Stochastic modelling of waiting time to the emergence of a resistant clone in current experimental approaches shows that this is extremely variable (re-plating every 2 weeks, 1:10, resistant mutation rate $2 \times 10^{-8}$, $10^4$ simulations). **c** Systems accommodating large populations are more likely to contain pre-existing resistant subclones, can be grown without re-plating, and lead to evolutionary dynamics of resistance that can be largely deterministic, reproducible and predictable. **d** Summary of the differences between current approaches and the approach we propose. **e** The selective effect of a drug on a heterogeneous population can be visualised as a fitness landscape. Genetically distinct cells are represented by points in the horizontal plane, whilst their fitness within a certain environment is the vertical axis. Different drugs have different landscapes, selecting against different clones. For simplicity, here we assume in the absence of drug all clones to be equally fit (flat landscape). Drug 1 changes the landscape, selecting for $y$ and $z$ but against $x$. Drug 2 selects only for $z$ and drug 3 only for $y$. **f** First, a population of cells is present at baseline, represented here as equally fit for simplicity. **g** Drug 1 selects clones that are resistant to drug 2. **h** Applying first drug 3 leads to evolutionary steering of a population that is entirely sensitive to drug 2. Here genotypes with values below the plain have negative fitness and so their frequency will decrease until they go extinct.

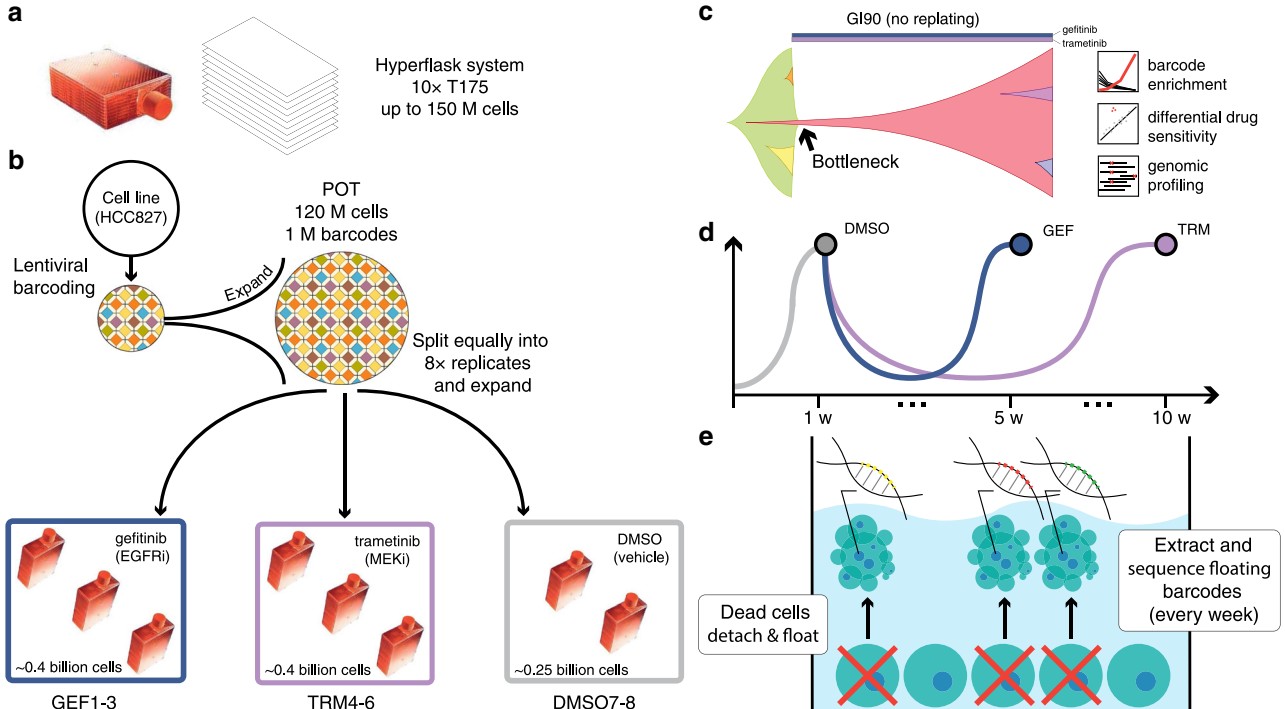

**Fig. 2 Experimental design. a** The Corning® High Yield PERformance Flasks (HYPERflask®) cell culture vessel is a 10-layer 1720 cm² total growth area system with polystyrene gas permeable surface that can reach the order of 150–200 million cells. **b** One million HCC827 lung cancer cells were lentivirally barcoded before being expanded to a population (POT) of ~120 million cells in a HYPERflask. Eight replicates were seeded with ~12 million cells each and expanded to 120 million. The remaining POT cells are frozen for subsequent analysis. Of the eight seeded replicates, three are exposed to GI90 doses of gefitinib (GEF1–3), three to GI90 of trametinib (TRM4–6) and two were used as controls (DMSO7–8). **c** Clonal evolution of a large population containing pre-existing resistant subclones that is exposed to high drug concentration (GI90) without re-plating. A clonal bottleneck occurs by means of Darwinian selection for drug resistance. Barcode enrichment analysis, genomic profiling and drug screening is performed on the resistant population. **d** Schematic growth curves for gefitinib (4 weeks for resistant population to regrow) and trametinib (9 weeks for the resistant population to regrow). Media and drug are changed weekly. **e** When cells die, they detach and float in the media. At each media change (once per week), supernatant cells are harvested from the spent media and their DNA extracted for barcodes analysis and non-destructive tracking of tumour evolution.

cancer cells that is the substrate for Darwinian selection to operate. When drug 1 is applied (Fig. 1g), only populations that are around the new fitness peaks survive, while drug-sensitive cells in fitness valleys go extinct. If then we expose the population to drug 2, which has an overlapping fitness peak, we select for a doubly resistant phenotype *z*, against which both drug 1 and 2 are ineffective. At this point we would have lost control of the tumour. Instead, if we first apply drug 3 (Fig. 1h) this leads to selection for phenotype *y*. Because drug 2 shows differential fitness peaks with respect to drug 3, the sequence drug 3–drug 2 leads to an evolutionary trap in which the cancer cell population goes extinct[31]. This is the principle of evolutionary steering that can be exploited to delay and potentially control drug resistance, thus significantly extending patient survival.

**Evolving resistance in large populations without re-plating.** We demonstrate evolutionary steering using the HCC827 non-small cell lung cancer line. HCC827 is an *EGFR* exon19del mutant lung cancer cell line sensitive to *EGFR* inhibition[40]. We chose HCC827 because it is a well-characterised line for which some mechanisms of resistance to EGFR inhibition are already known, such as pre-existing *MET* amplification[40]. We used two small molecule inhibitors for steering: gefitinib, an *EGFR* inhibitor, and trametinib, a *MEK1/2* inhibitor. To recapitulate the evolutionary dynamics of large populations, we employed a HYPERflask® cell culture system, wherein each flask has a capacity of up to 150–200 million cells, about 10 times higher than a normal T175 flask (Fig. 2a). To track clonal evolution we employed high complexity

lentiviral barcoding[41], a now established technique to study drug resistance[35,42]. By barcoding the cells at baseline and splitting them into distinct replicates (Fig. 2b), we could determine whether resistant clones were pre-existing if the same barcodes were enriched post-treatment in different replicates. We first barcoded a population of one million cells with one million distinct barcodes, and then expanded it to ~120M in a HYPERflask (see "Methods" section). We call this initial baseline population the "POT" (Fig. 2b). For each of the two drugs we seeded three HYPERflask replicates in addition to two HYPERflask as DMSO controls. Each HYPERflask was seeded with ~15 million cells from the same POT (i.e. most barcodes are common to all flasks) and expanded to 80–90% confluence. Thus, we achieved a total population of $120 \times 10^6 \times 3 = ~0.4$ billion cells per drug arm (Fig. 2b). Stochastic mathematical modelling demonstrates that this experimental design leads to each replicate being representative of the POT (see "Methods" section and Supplementary Fig. 2).

These large populations allowed us to expose the cells to high drug concentrations without causing extinction and without the need for re-plating. This is because large populations are highly heterogeneous and likely to contain pre-existing resistant subclones that would survive high-dose drug exposure. We used GI90 concentrations (90% Growth Inhibition) until resistant clones grew back (Fig. 2c and Supplementary Fig. 3). Three HYPERflasks were drugged with gefitinib (40 nM) and three with trametinib (100 nM). Drug exposure in the gefitinib-treated lines GEF1–GEF3, induced extensive cell death, causing a major

population bottleneck. Under constant drug concentration, the resistant population grew back and reached confluence again in 4 weeks. Drug exposure in the trametinib-treated lines TRM4–TRM6 also induced extensive cell death and a resistant population grew back to confluence in 9 weeks (Fig. 2d).

We reasoned that not only the surviving resistant cells at the end of the experiments were important for the analysis, but also that the cells that died during the experiment could prove informative on the temporal dynamics of the system. The idea is that the sum of the surviving cells attached to the plate and the dead cells floating in the media would contain information on the whole evolutionary history of the cell population. Moreover, we hypothesised that dead cells may be a representative sample of the live population and, like circulating tumour DNA in cancer patients[43], could be used to monitor the temporal dynamics of the system in a non-destructive way. Once a week at each media change, we collected the floating (dead) cells as pellets to extract DNA and perform barcode analysis (Fig. 2e, see "Methods" section).

We compared baseline (POT) vs. resistant lines and confirmed decreased drug sensitivity for both gefitinib (Fig. 3a) and trametinib (Fig. 3b). To identify possible genetic mechanisms of resistance, we performed whole-exome sequencing at 160× median depth. We found a focal amplification of *MET* in gefitinib-resistant lines (Fig. 3c), consistent with previous results[40] and that was confirmed by digital droplet PCR (ddPCR) (Supplementary Fig. 4). No amplification of *MET* was detected in trametinib-resistant lines, suggesting that *MET*-amplified sub-clones are gefitinib-resistant but may be trametinib-sensitive. The trametinib-resistant lines shared a gain of chr1p and deletions in chr9, encompassing *CDKN2A* (Figs. 3c, S4, S5, S6 and Supplementary Data 1). *CDKN2A* encodes tumour suppressors p16 and p14ARF and loss of this gene has been linked to resistance to targeted drugs[44], although never in the context of trametinib resistance. Analysis of single nucleotide variants (SNVs) revealed a small cluster of mutations clearly enriched in the trametinib-resistant lines compared to POT (Fig. 3d, S7). These mutations were also enriched in the gefitinib-resistant lines, although to a lesser extent, potentially indicating a pre-existing subclone that is doubly resistant to gefitinib and trametinib, although more strongly selected by trametinib.

The fact that genomic alterations were consistent between evolved replicates but different for the two drugs suggests that multiple resistant subclones were already present in the initial population. Differential evolution and competition of these subclones under the two drugs also suggests a target for steering.

**Tracking clonal evolution in real time non-destructively.** We next sought to more precisely quantify the temporal evolutionary dynamics of drug resistance. We profiled the barcodes of all samples using ultra-deep sequencing (see "Methods" section and Supplementary Fig. 8). In comparison to the 2295 unique barcodes identified in the POT population, we found an average of 872 unique barcodes in the gefitinib-treated lines and an average of 199 unique barcodes in the trametinib lines (Supplementary Fig. 8), indicating that drug exposure induced a strong selective bottleneck. We note that because of the single-cell barcoding, we expect multiple barcodes corresponding to each pre-existing subclone (i.e. multiple cells in the subclone have been barcoded with different barcodes). We shall also note that the number of unique barcodes identified with deep sequencing in the POT is a small subsample of the total number of barcodes present in the sample due to limitations of sequencing which can only identify haplotype frequencies as low as 1–0.1%. We sequenced at a depth of 300,000–600,000× (see Supplementary Fig. 8B), and hence

considering that we start with ~150 cells per barcode, we can only pick up at most some thousands of unique barcodes.

We considered a barcode as positively selected in a given replicate when its estimated growth rate was positive with respect to DMSO (see "Methods" section). We grouped barcodes with similar growth dynamics into 'functional subclones'. We define pre-existing functional subclones as those having similar growth dynamics in more than one replicate (Fig. 4a and see "Methods" section for details). Notably, we cannot exclude that each functional subclone may be composed of multiple genetically distinct subclones. This is not critical for our analysis as we are interested in drug response phenotypes, rather than individual genotypes.

We identified five functional subclones with different growth dynamics (Fig. 4b). The first group (grey) was the largest (87.2%) and represented largely clones that died under both drugs (sensitive) as well as clones for which the growth rate could not be determined because not found in the DMSO (Supplementary Fig. 9). The second group (blue) was resistant to gefitinib but sensitive to trametinib. The third group (purple) was resistant to trametinib but sensitive to gefitinib. The fourth group (orange) was doubly resistant to both drugs. Finally, the fifth group (green) was composed by a set of barcodes that were found in only one replicate either of trametinib or gefitinib. This set could correspond to possible de novo resistant lineages. As this group comprises barcodes at very low frequency, we focused on the majority of pre-existing resistant subclones that are relevant to evolutionary steering. We examined the frequency of barcodes and associated phenotypes in the POT versus the evolved lines. Strikingly, the frequencies of barcodes between replicates of a drug were highly similar, confirming that the initial conditions are a strong determinant of evolution under exposures to high drug concentrations (Fig. 4b). Importantly, these results indicate that in this system, dynamics are largely deterministic and hence predictable.

We reasoned that the doubly resistant (orange) subclone could be the one carrying the SNVs found highly enriched in TRM and partially enriched in GEF using the exome sequencing analysis. We contrasted the barcodes frequency of the orange subclone with the SNV cancer cell fraction (CCF) in each sample and found that these two independent measurements matched in all samples, including the POT, thus suggesting that those SNVs and the orange barcodes are in the same cells (Fig. 4c).

Using mathematical modelling, we measured the growth rates of each barcode under each condition (see "Methods" section). This analysis confirmed that gefitinib-resistant population was polyclonal, with a large *MET*-amplified subclone (blue barcode group) composing ~32.8% (average) of the population in GEF1–GEF3 and a relatively large initial population (~2.4%) in the POT (see Fig. 4b). This subclone was characterised by many barcodes with a positive growth rate under gefitinib but a negative growth rate under trametinib (Fig. 4d—blue barcodes). We also found enrichment for the multidrug-resistant subclone (orange barcodes) that exhibited a positive growth rate under both gefitinib and trametinib. This subclone was found at mean frequency of 22.4% in the GEF lines and 86.1% in the TRM lines (Fig. 4d—orange barcodes). This clone was smaller than the blue clone in the original POT population (~0.91%) and therefore carried many fewer barcodes. There was also a small set of barcodes that were only enriched in the trametinib lines (Fig. 4d—purple barcodes, ~4.2% average in TRM lines, ~0.57% in POT). The combined frequency of all enriched (resistant) barcodes, belonging to distinct subclones in the initial population, was 3.9%. The growth rates across replicates were highly similar (Fig. 4e and Supplementary Fig. 10). Hence, the barcode analysis supports the presence of pre-existing polyclonal drug resistance.

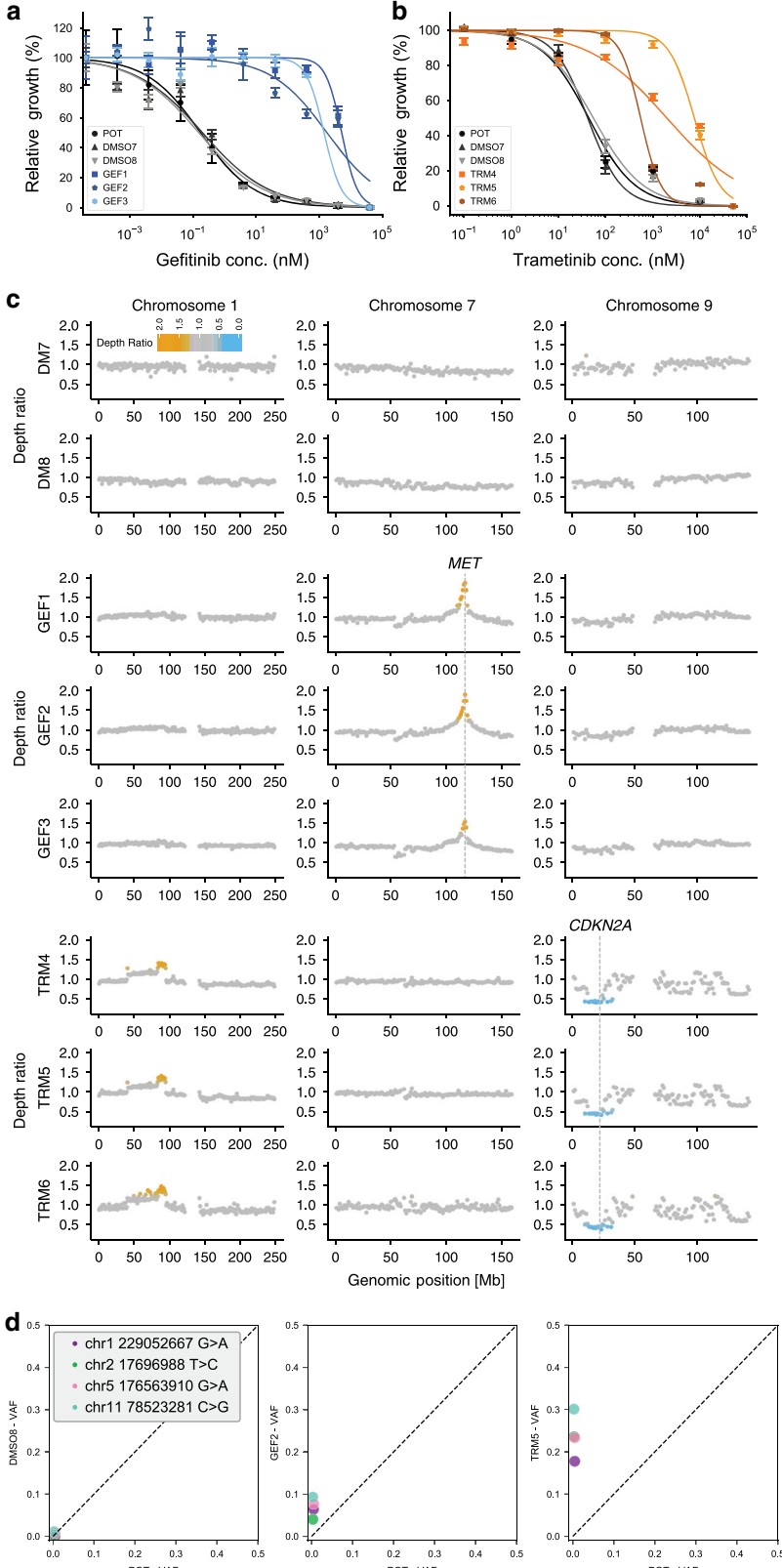

**Fig. 3 Characterisation of resistant lines. a** Dose–response curves of gefitinib-resistant lines and **b** trametinib-resistant lines versus DMSO demonstrate acquired resistance. Error bars of dose–response curves represent SEM. **c** Relative copy number profiles of resistant lines compared to POT highlight *MET* amplification in GEF lines and 1p gain, 9p loss (including *CDKN2A*) in TRM lines. **d** Single-nucleotide variant analysis shows enrichment of a subclone containing four variants in TRM and partial enrichment in GEF of the same clone (VAF variant allele frequency).

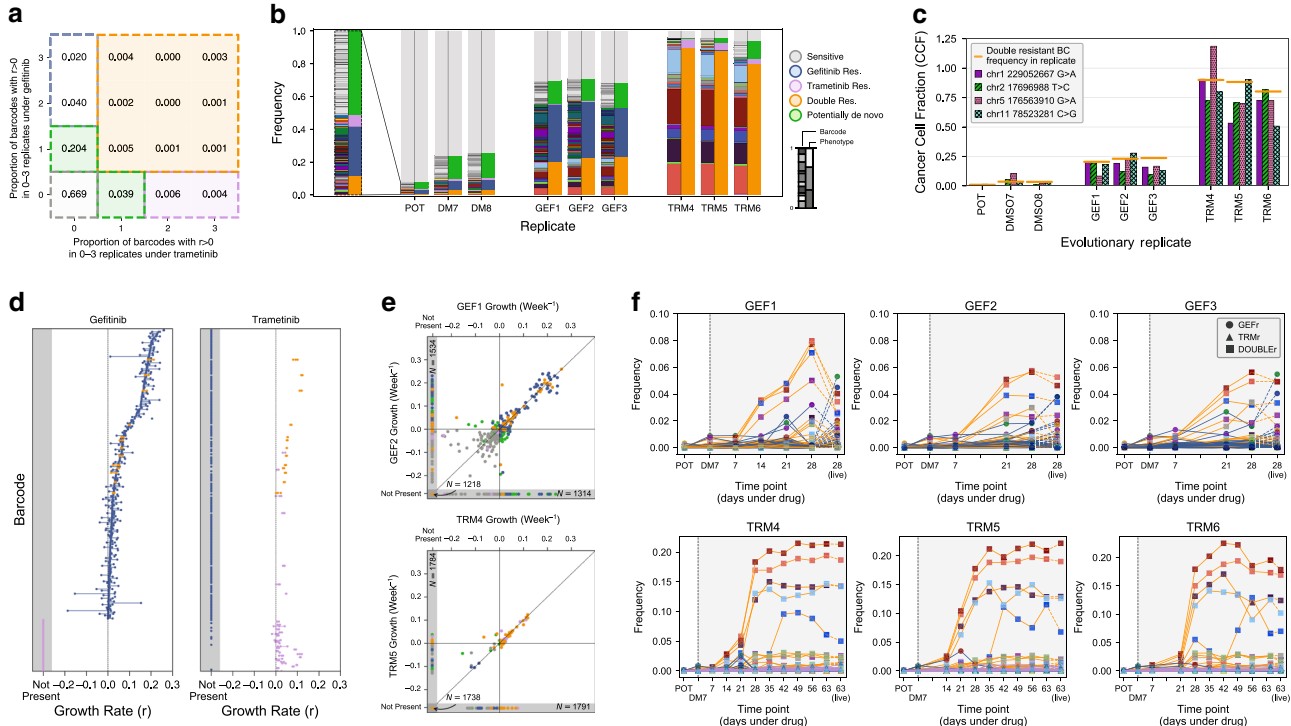

**Fig. 4 Barcodes reveal evolutionary dynamics over time. a** Map of barcodes associated to drug response shows proportion of barcodes that were sensitive to both drugs (in grey—bottom left corner), versus resistant to gefitinib only (blue), trametinib only (purple) or both (orange). Values indicate the proportion of unique barcodes with positive growth rates across the given number of replicates of gefitinib exposure and trametinib exposure. **b** Barcode frequency distributions in each sample. Left hand bars show the frequency of each unique barcode. Barcode colours and ordering are identical between replicates. Right hand bars indicate the drug response phenotypes assigned to each barcode (see right hand legend). Phenotypes are determined by the set of evolutionary replicates in which a barcode exhibits a positive growth rate (see "Methods" section). Barcode and phenotype distributions are highly conserved between replicates, indicating repeatable evolution. **c** Cancer cell fraction estimates for the cluster of four SNVs identified form exome sequencing matched barcode frequencies for the doubly-resistant clone. **d** Growth rates for each barcode assigned to the GEF, TRM or double resistant phenotypes are shown under both the gefitinib (GEF1–3, left) and trametinib (TRM4–6, right) exposure. Points indicate the growth rates in the three replicates and lines connect these points to highlight variance. **e** Representative scatter plots show the concordance in barcode growth rates between evolutionary replicates. Points are coloured according to barcode phenotype, as in **b**. **f** Temporal frequencies for the floating barcodes in each evolutionary replicate. Lines are coloured by phenotypes and marker colours correspond to the unique barcode as in **a**. POT and DM7 measurement are harvested (live) populations as is the final time point, all others are floating barcode measurements. The temporal frequency dynamics are conserved between time points of different replicates. Moreover, the final samples (harvested population) largely match the last floating barcodes samples.

As part of our experimental design, we never re-plated cells following drug exposure in order to avoid stochastic drift effects and sampling bias. As such, we could not take aliquots of cells for analysis throughout the experiment. To track evolution through time in a non-destructive way, we leveraged the large volume of media (560 ml) that is changed every week. HCC827 is an adherent cell line, with cells that detach from the plate surface upon death. We collected pellets consisting of cells that had died and extracted barcodes from each time point. We confirmed that pellets from supernatant collection were apoptotic/necrotic cells (Supplementary Fig. 11). Time-course barcodes allowed us to track the evolution under drug exposure without perturbing the system and at high resolution (Fig. 4f). This barcode analysis clearly showed an expansion of the subclones we identified in the final populations, with the final time point of barcodes derived from supernatant cells being very similar to the final harvested populations (Fig. 4f, line colours indicate phenotype, point colours indicate unique barcodes). This result seems partially counter-intuitive, as one might expect the barcodes harvested from the dead cells not to correspond to a resistant clone. However, this phenomenon can be understood by consideration of the under-lying evolutionary dynamics. At first, many barcodes are driven to extinction, because the majority of cells in the initial population

are sensitive to the drug. At this stage, those cells present in the harvested media correspond to the thousands of different barcodes of sensitive cells (grey), none of which is common in the initial population, hence no enrichment is detected. As the resistant population grows, the contribution to the floating media becomes a mixture of sensitive cells being driven to extinction, and resistant cells turning over. At the end of the experiment, it is these resistant cells that are dominant, with most floating cells (and therefore barcodes) representing the underlying resistant popula-tion dividing and turning over. The frequencies of the clones stabilised after ~3 weeks of gefitinib exposure, and 6 weeks of trametinib exposure. By comparing the time series barcode dynamics between replicates, we again saw that the temporal evolutionary dynamics are strikingly conserved, suggesting that the resistance dynamics are highly predictable (Fig. 4f).

**Single-cell analysis confirms pre-existing polyclonal resistance.** Genomic analysis revealed a *MET*-amplified clone in the gefitinib-treated lines and a separate CDKN2A-loss clone in the trametinib-treated lines. We performed single-cell RNA-seq on the POT sample, one gefitinib-treated replicate (GEF1) and one trametinib-treated replicate (TRM4). tSNE analysis confirmed that cells derived from the same sample clustered together

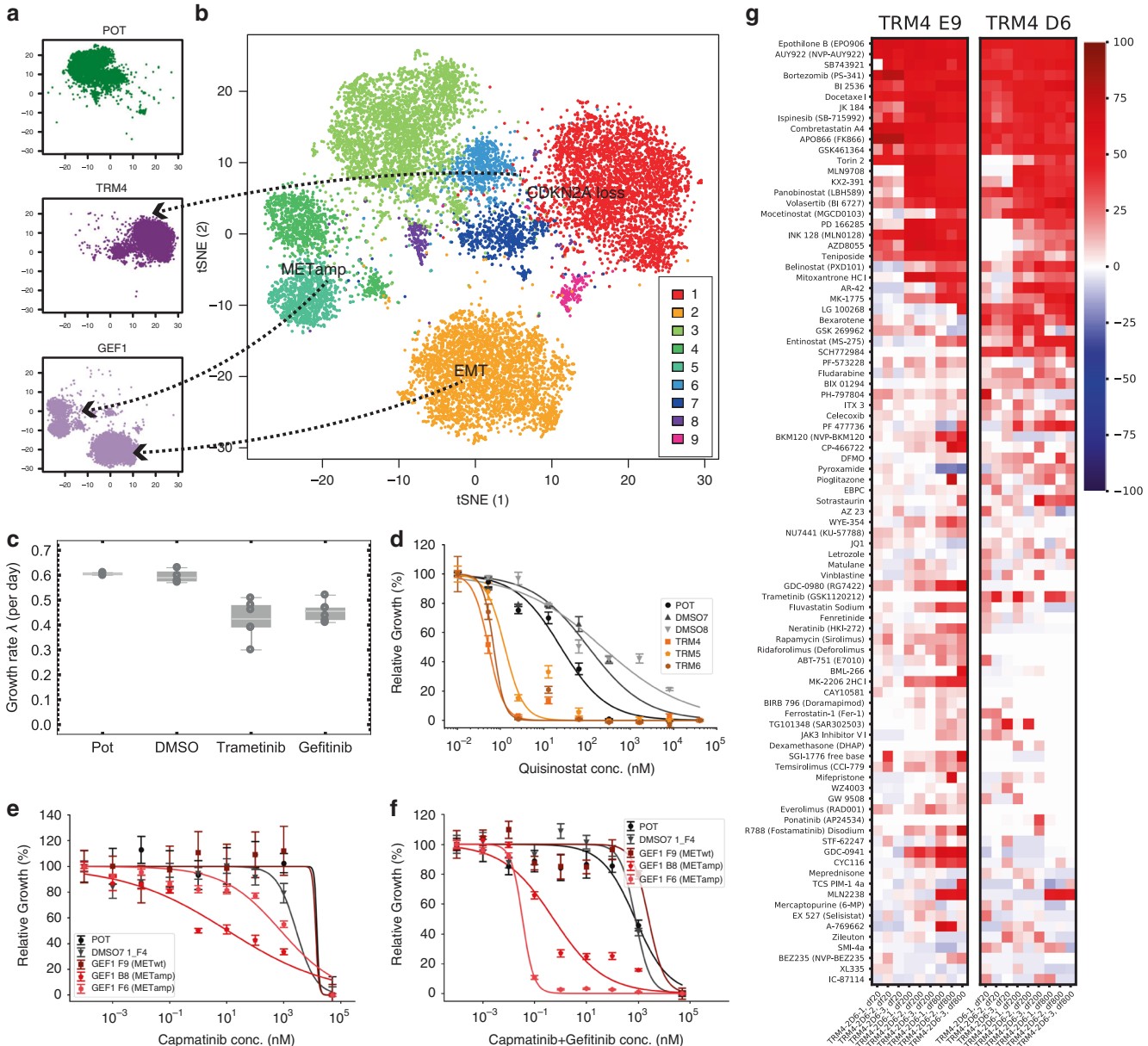

**Fig. 5 Evolutionary steering leads to collateral drug sensitivity. a** tSNE plot of single cell RNA sequencing coloured by sample (POT, TRM4 and GEF1 from top). **b** tSNE analysis of all samples together identifies 9 clusters with distinct transcriptomic profiles. **c** Growth rates of all lines in the absence of drugs show a cost of resistance in terms of proliferation in the evolve lines. Box plots show mean, interquartile values and the range. **d** Dose response curves of second generation HDAC inhibitor quisinostat show collateral drug sensitivity in TRM lines. **e** Dose response curve of single clones derived from gefitinib-resistant lines under *MET* inhibitor capmatinib show collateral sensitivity for clones that bear *MET* amplification (only B8 and F6). **f** Combination of capmatinib + gefitinib further increases collateral sensitivity of *MET*-amplified clones, in particular F6. **g** High throughput drug screening of 485 compounds shows collateral sensitivity in TRM clones TRM4 E9 (1 copy left of *CDKN2A*) and TRM4 D6 (two copies left of *CDKN2A*). Here we show top 5% with highest % change inhibition with respect to DMSO7 F4 (three copies of *CDKN2A* as HCC827 is a triploid line). Error bars of dose–response curves represent SEM.

(Fig. 5a). Gene expression patterns confirmed that the gefitinib-resistant population (GEF1) was composed largely of two major subclones, one that was *MET* amplified (Fig. 5b, clusters 4 and 5) and one that was EPCAM−/VIM+, indicative of epithelial to mesenchymal transition[45] or EMT (cluster 2). Indeed, EMT has been implicated in gefitinib resistance in lung cancer[35]. In agreement with the pre-existing nature of these subclones, over-expression of *MET* was detected in a subset of cells in the POT (cluster 3—see Supplementary Figs. 12 and 13). An *EPCAM−/VIM+* subpopulation was also present in the POT (cluster 9—see Supplementary Figs. 12 and 13). The trametinib-resistant

population was mostly composed of a single *CDKN2A* loss subclone (Fig. 5b, cluster 1) and again *CDKN2A* loss was detectable in a subpopulation of cells from the POT (cluster 6—see Supplementary Figs. 12 and 13). Hence, the scRNA-seq data confirms the clonal composition reported by the barcodes. No over-expression of P-glycoprotein (PGP), a known multidrug resistance gene, was detected (Supplementary Fig. 13), supporting the idea that drug-resistant clones have heritable phenotypes. Phosphoproteomic results validated our findings in terms of the functional effects of the drugs on the signalling pathways (Supplementary Fig. 14, see "Methods" section).

**Evolutionary trade-offs and collateral drug sensitivity.** We wanted to determine whether drug adaptation came at a cost in terms of proliferation rate or increased sensitivity to a second drug. We measured the growth rates of the evolved lines in the absence of drug and confirmed that they all grow significantly slower than the parental line and DMSO controls (Fig. 5c and Supplementary Fig. 15). Hence, a cost in terms of proliferation that is now encoded in the genotype of the population has occurred. This evolutionary trade-off can potentially be exploited using a form of adaptive therapy termed 'buffer therapy'[24], where the sensitive population can be used to keep in check the resistant population through competition.

We then sought to test for collateral drug sensitivity to other compounds that have shown evidence of being effective in *EGFR* mutant NSCLCs. Histone deacetylase (HDAC) inhibitors are a new class of drugs that have shown promising results in NSCLC patients, and to which HCC827 is known to be sensitive[46,47]. We did identify collateral sensitivity for second generation HDAC inhibitor quisinostat in the trametinib-resistant lines, which displayed <200× IC50 with respect to DMSO (Fig. 5d) but not for pan-HDAC inhibitor Panobinostat (Supplementary Fig. 16A). Another class of new promising drugs for NSCLCs are Aurora A Kinase inhibitors[48]. Aurora kinases have also shown to drive resistance to third generation EGFR inhibitors[49]. However, we found no collateral sensitivity for those inhibitors in our lines (Supplementary Fig. 16B, C).

We then reasoned that the gefitinib resistant subclones may exhibit collateral sensitivity to *MET* inhibition with capmatinib. However, sensitivity was not increased in the bulk gefitinib resistant population (Supplementary Fig. 16D). As both barcodes (Fig. 4b) and single-cell transcriptomics (Fig. 5b) indicated polyclonal resistance in the GEF population, with a mixture of *MET* amplified and non-amplified clones, we isolated individual single-cell-derived subclones from DMSO and resistant lines and confirmed which ones were *MET* amplified and WT by ddPCR (Supplementary Fig. 17A). We did the same for the trametinib resistant line and verified *CDKN2A* loss amongst the isolated clones (Supplementary Fig. 17B). Indeed, individual *MET*-amplified clones, such as G1_B8 and G1_F6, showed increased sensitivity to capmatinib, with >255× lower IC50 for G1_B8 and >3.5× lower IC50 for G1_F6 with respect to DMSO (Fig. 5e). We hypothesised that, given that both clones harboured >13 copies of *MET*, the difference in increased sensitivity in G1_F6 was due to residual *EGFR* signalling compensating for *MET* inhibition by capmatinib in this clone. We tested for collateral sensitivity to the combination of gefitinib + capmatinib, and indeed achieved >23,000× lower IC50 in G1_F6 than DMSO (Fig. 5f). We speculate that the decreased sensitivity to the combination of capmatinib + gefinitib of POT may be due to some antagonism between the two drugs.

For trametinib, as *CDKN2A* loss leads to upregulation of *CDK4/6*, we reasoned that inhibition of CDKs could prove effective against the trametinib-resistant population. Although *CDK4/6* inhibitor palbociclib alone was not effective (Supplementary Fig. 16E), combination of palbociclib + trametinib showed some level of sensitivity for the clone with the highest loss of *CDKN2A* (clone TRM4 E9—see Supplementary Fig. 17B), with IC50 reduced by 14× with respect to DMSO (Supplementary Fig. 16F). To scale up our search for collateral sensitivity to trametinib, we leveraged on high throughput drug screening technology to assay a panel of 485 compounds (see "Methods" section) at each of three concentrations (20, 200 and 800 nM) with three replicates of clones TRM4 D6 and E9 vs. DMSO7 F4. This screen revealed a large number of collaterally sensitive compounds. The 5% of compounds with the highest % change in inhibition is reported in Fig. 5g.

## Discussion

The vast majority of metastatic cancers remain largely incurable. Treatment with standard approaches may extend survival[1], but ultimately fails due to the emergence of resistant cells[4]. This is the natural consequence of a process of clonal evolution fuelled by ITH[10]. Combining different drugs together at the same time has been investigated, but typically only improves survival by a few months, if any[50,51], and the narrow therapeutic window of cancer drugs leads to high toxicity in combinations, limiting the practicality of this approach. Instead, controlling the disease, rather than attempting to cure it, may be the only viable option in advanced cancers[24]. Although this sounds radical in oncology, resistance management is well established in fields such as HIV[52], antibiotics[19] and pest control[53–55]. In cancer, different groups have explored the concept of 'adaptive therapy', first pioneered by Gatenby[24], where drug dose is modulated in response to the underling evolutionary dynamics[56,57], with encouraging preliminary results in clinical trials[58]. Many adaptive approaches are based on 'buffer therapy', which exploits the fact that resistance often comes at a proliferative cost and hence resistant subpopulations may be outcompeted in a drug-free environment[27]. This evolutionary double bind[25,26] has been observed prospectively in colorectal cancer patients under *EGFR* inhibition, where *KRAS*-driven resistance seems to imply a cost, and *KRAS* subclones decrease in relative frequency if the drug is suspended[28]. We have also observed this in our evolved lines treated with trametinib, which show significantly slower growth with respect to baseline. When resistance comes at a cost in a drug-free environment, the drug-sensitive subpopulations can be used to "keep in check" drug-resistant cells[24]. This would explain the low prevalence in the POT of the *CDKN2A*-loss and *MET*-amplified clones. Moreover, evolutionary game theory has been proposed as a conceptual framework for adaptive therapy, in which cellular phenotypes are represented as strategies in a game[59]. In light of these advances in adaptive therapy, in this study we evaluated evolutionary double binds that could be exploited with evolutionary steering to control or prevent drug resistance.

Despite the conceptual elegance and promises of adaptive therapy however, current strategies are often based on ad hoc rules of thumb. The lack of reliable experimental model systems that recapitulate patient heterogeneity and clonal evolution is a major barrier for bringing adaptive therapies to the clinic. Here we presented an approach for clonal steering where evolution can be tightly controlled, monitored and altered using drugs. This has the potential of paving the way to multidrug adaptive treatments.

Although we have attempted to design a model system that specifically aims at recapitulating the evolutionary dynamics of treatment resistance occurring in patients, our study has limitations. First, we do acknowledge that an established cell line with a clonal oncogenic driver in *EGFR* may not recapitulate the dynamics of evolutionary steering in patients. Second, we have used high concentrations of drugs that may not be always achievable in patients. Third, future studies will be needed that incorporate tumour microenvironment factors, such as cancer-associated stromal and immune cells, as well as different doses of drugs. Fourth, here we focus on the study of drugs as selective pressure for pre-existing resistant clones but we do acknowledge the importance of de novo mutants as well. For example, in a clonal cell line with no pre-existing resistance one would expect that all resistance dynamics are driven by drug tolerance followed by de novo resistance. This phenomenon is often described by bet-hedging dynamics[12]. This could be potentially studied with the presented platform, although one would expect different barcodes in different replicates, making the evolution highly stochastic. On the other hand, the floating barcodes would allow us to determine the waiting time for a de novo mutant with

great precision, and hence the measurement of temporal dynamics in the context of bet-hedging, which are key to understanding mutation rates and the dynamics of resistance. In conclusion, additional validation experiments will be needed prior to the adoption of this type of framework into clinical trial design. The first step would be to apply this framework to patient-derived organoid models, which been shown to recapitulate clinical outcome[60].

Despite these limitations, model systems that replicate the temporal dynamics of human cancer evolution will shed new light on how to control drug resistance in advanced cancers, and open the opportunity of personalised adaptive drug schedules that may achieve long-term control in advanced human malignancies.

## Methods

**Mathematical modelling of experimental evolution approaches.** In order to get the expected distribution of waiting times for the occurrence of resistant mutations in typical in vitro re-plating experiments we did individual-based stochastic simulations of the original cell growth and cell sampling process. We start the simulation with $2 \times 10^6$ non-resistant cells. Cells are randomly picked for division and the population is grown to a size of $4 \times 10^7$ cells, which corresponds to the expected cell population size after 14 days with an average cell division rate of once every 3 days. During each division cells hit a resistance inducing mutation with probability $\mu = 2 \times 10^{-8}$. Given a healthy mutation rate of $1 \times 10^{-9}$ bp/cell division this approximately implies 20 different resistance inducing mutations. Once the population reaches $4 \times 10^7$ cells, cells were replated, which in our simulation corresponds to a population size reduction to $2 \times 10^6$ cells. The growth and resampling process was repeated until the first resistance inducing mutation occurred. We ran $10^4$ independent stochastic simulations and recorded the times of resistance occurrence, which allowed us to construct the expected distribution of waiting time. In Fig. 1d we estimated the expected number of mutants arising in an expanding population in different scenarios, using:

$$E(\# \text{ of mutants}) = \mu (N_{max} - N_0) \times \text{replatings}$$

where $\mu$ is the mutation rate of the resistant mechanism, $N_0$ is the seeded population in the flask/well and $N_{max}$ is the maximum number of cell capacity before re-plating.

To demonstrate that each of the 8 HYPERflask is representative in terms of barcodes, we performed a stochastic population simulation of the whole splitting and growing step to estimate the likelihood that a barcode in an initial population is present in $N/8$ of the replicate populations. The simulation comprised of two parts:

a. Stochastic simulation of the POT outgrowth from an initial population of uniquely barcoded cells.
b. Stochastic simulation of splitting the POT population into eight replicate populations.

To achieve (a) we assumed that each cell in the initial population was uniquely barcoded, and that each uniquely barcoded population was subject to stochastic exponential growth with birth rate $b$ and death rate $d$. We implemented a Gillespie algorithm to simulate the exponential growth (Supplementary Fig. 2A, see ref. [61] for details). Birth and death rates for the HCC827 cell line were previously derived in ref. [62]. For the oxygen concentration of 20% and media glucose concentration of 2 g/L that correspond to our experimental design, the appropriate values are approximately $b = 0.032$, $d = 0.002$, which we used to parameterise the model. Supplementary Fig. 2B shows a histogram of population sizes from 10,000 realisations of the simulation from a single cell with instances of extinction (population size equals zero) omitted.

Under this model of stochastic exponential growth differently barcoded populations do not interact, and so the POT barcode frequency distribution was computed by combining 10,000 independent realisations of the stochastic process. The barcode frequency distribution that arises is shown in Supplementary Fig. 2C.

Finally, to simulate (b) we performed a random equal size 8 way split of the full population of barcodes generated by the stochastic simulation. To determine the likelihood that a barcode appears in precisely $N/8$ replicates, we simulated the stochastic outgrowth of the POT 10 times, each with 20 associated stochastic simulations of the split, and averaged the results. The predictions are shown in Supplementary Fig. 2D. We find that ~90% of the barcodes that survive the POT outgrowth appear in 8/8 replicates, with an additional 4% and 2% appearing in 7/8 and 6/8, respectively. Approximately 0.01% of barcodes appear in precisely one replicate.

This calculation is indeed confirmed by the fact that all replicates contain a statistically similar repertoire of barcodes comes from the fact that in all three replicates exposed to trametinib and all three replicates exposed to gefitinib showed enrichment for exactly the same set of barcodes that were rare in the POT. Hence, all rare barcodes are well represented in each replica. See Supplementary Methods for additional statistical analysis of barcodes.

**Analysis of growth curves.** HCC827 cells from POT, DMSO7, DMSO8, GEF1–3 and TRM4–6 were seeded at a density 1000 and 3000 per well in a 96-well and 48-well plates (Corning), respectively. Cells were grown in six independent replicates in 96-well plate and three independent replicates 48-well plates for the duration of total 8 days. Media containing neither a vehicle control nor a drug were used as a fresh media every 3 days. The plates were placed into the IncuCyte® S3 Live-Cell Analysis System (Sartorius) and images were taken every 2 h at ×4 and ×10 magnifications. At the end of 8 days, confluency determination for each of the time points were automatically calculated based on the images acquired using IncuCyte® S3 Live Cell Analysis System (Sartorius). We calculated the growth rate of each line using linear fitting of log-transformed data (see Supplementary Fig. 15).

**Cell line culture in HYPERflasks.** HCC827 cell line was cultured in RPMI-1640 medium (Sigma-Aldrich) supplemented with 10% FBS (Sigma-Aldrich), 4 mM L-glutamine (Sigma-Aldrich), 1% non-essential amino acids (Sigma-Aldrich), and 1% penicillin–streptomycin (Sigma-Aldrich). Cell line was confirmed to be Myco-plasma free using PCR-based method. Cell line was grown and expanded in High Yield PERformance Flasks (HYPERflask®) cell culture vessel (Corning). Medium was changed once a week and cells were harvested upon reaching ~85% confluence.

**Barcoding of cell lines.** The ClonTracer lentiviral barcode library construction and the generation of the lentivirus were previously described (38). The ClonTracer library was a gift from Frank Stegmeier (Addgene #67267). HCC827 cell lines were cultured in normal growth media and barcoded by lentiviral infection using 0.8 µg/ml polybrene. For the majority of single cells to be infected with a single barcode a multiplicity of infection (MOI) of 0.1 corresponding to 10% infection was chosen, following lentiviral titration results. Following infection, 2.5 µg/ml puromycin was used for selection of cells infected with a barcode. Statistical analysis of the bar-coding process suggests that <1% of cells were doubly barcoded and <0.1% of unique barcodes were received by multiple cells (Supplementary Methods).

**Generation of gefitinib and trametinib-resistant cell lines.** The 1 million bar-coded HCC827 cells were expanded to ~120 million cells, harvested and frozen. Of these frozen cells, 4 million cells were thawed and again expanded to ~120 million cells. These cells were seeded into eight HYPERflasks equally. Two HYPERflasks were grown under <0.0001% of DMSO for 6 days upon which they reached ~85% confluence and were harvested as controls. The remaining six HYPERflasks were grown under normal growth media for 1 week upon which they were exposed to GI90 concentrations of gefitinib (Selleckchem) and trametinib (Selleckchem) (three replicate flasks for each), for 4 and 9 weeks, respectively. During this time, the medium and inhibitor were replenished weekly. The GI90 concentrations for gefitinib and trametinib were previously determined to be 40 and 100 nM (Sup-plementary Fig. 1). Cell counts were determined via the Countess II Automatic Cell Counter (ThermoFisher).

**Barcode amplification and next generation library preparation.** Barcoded HCC827 cell lines were harvested and pelleted. Genomic DNA isolation was performed using DNeasy Blood and Tissue DNA extraction kit (Qiagen) according to manufacturer's recommendations. Half of the conditioned media from each HYPERflask was centrifuged at 1,200 rpm and pelleted. Quantification of genomic DNA was carried out using Qubit (Life Technologies). Amplicon PCR reaction was performed using 2x Accuzyme mix (Bioline) and 20 ng of DNA to amplify the barcode using the previously published primer sequences[41]:

Forward: ACTGACTGCAGTCTGAGTCTGACAG.
Reverse: CTAGCATAGAGTGCGTAGCTCTGCT.

Following detection of 80-bp PCR product including the 30-bp semi-random barcode and after purification, NGS libraries were prepared using the NEBnext Ultra II DNA library preparation kit for Ilumina (New England Biolabs) according to manufacturer's recommendations. Libraries were quantified using Qubit (Life Technologies) and KAPA library quantification kit (KAPA Biosystems), as well as TapeStation (Agilent Genomics). Library preparation was not successful for DNA extracted at four floating cell time points (GEF2-F2, GEF3-F2, TRM5-F1 and TRM6-F3). NGS was performed in house using MiSeq (Ilumina).

**Barcode bioinformatics analysis.** FastQ files were first filtered to extract those reads with quality score >20 in all positions. Reads matching potential barcodes were extracted from FastQ files by use of a regular expression matching 12 bases of the forward barcode primer, followed by 30 base pairs, followed by 12 bases of the reverse barcode primer. To account for potential errors arising from PCR ampli-fication or mutation, similar barcodes were merged via a method (outlined in the Supplementary Methods) that assigns each barcode to a representative matching the known weak/strong base pair pattern by consideration of the Hamming dis-tance between barcodes. Barcode frequencies are reported in Supplementary Data 2.

To assign a phenotype to each barcode we first determined an approximate growth rate under each condition by consideration of the frequencies. We assumed that the frequency of each barcode in the DMSO replicates was representative of the frequency in the drug-treated replicated GEF1–GEF3, TRM4–TRM6 prior to

the introduction of drug. To ensure a conservative estimate of the growth rate, we estimated the initial frequency as $f_0 = \mathrm{Max}(f_{D7}, f_{D8})$ where $f_{D7}, f_{D8}$ denote the frequency of the barcode in the lines DMSO7 and DMSO8, respectively. Denote the barcode frequency in a given replicate following drug exposure, expansion and harvesting by $f_R$. We estimated the growth rate of the barcode under drug exposure as

$$r = \frac{1}{T}\log\left(\frac{f_R}{f_0}\right)$$

where log denotes the natural logarithm and $T$ denotes the time between drug exposure and harvesting the cells ($T = 4$ weeks for gefitinib, $T = 9$ weeks for trametinib).

Phenotypes were then assigned according to the number of gefitinib and trametinib evolutionary replicates in which the barcode exhibited a positive growth rate. As a barcode can appear extinct in a given replicate either because it has negative growth rate, because the specific barcode was never seeded to that replicate, or because of drift, we determined barcode phenotypes as follows. Where a barcode exhibited positive growth rate in both 1+ GEF and 1+ TRM replicates, the barcode was designated as double-resistant. Where a barcode exhibited positive growth rate in 2+ GEF lines but no TRM lines, it is designated gefitinib resistant/ trametinib sensitive. Likewise, where a barcode exhibited positive growth rate in 2+ TRM lines but no GEF lines, it is designated trametinib resistant/gefitinib sensitive. Where a barcode exhibits positive growth rate in a single replicate (GEF or TRM), it is designated as putatively de novo resistance. Other barcodes with measured growth rate are designated sensitive. Finally, some barcodes are designated as having undetermined phenotype where a barcode is not detected in DMSO7 or DMSO8 (potentially due to loss at seeding) but observed in a replicate, as a growth rate cannot be determined. Figure 4a shows a schematic of the phenotype mapping along with the proportion of unique barcodes assigned to each phenotype. Moreover, we compared a previous 'POT' baseline sample with the POT used in this experiment, after it has been frozen, stored and then thawed. Supplementary Fig. 14 shows that barcodes are highly consistent in terms of proportion in the two samples, with a proportion of barcodes that are always missed by sequencing, which implies a binomial sampling of the barcodes.

**Whole exome sequencing**. Nine whole exome sequencing libraries were prepared from 200 ng of genomic DNA using the Agilent SureSelect HT2 Human All Exon_V6 kit following the manufacturer's instructions. The libraries were pooled and sequenced on the Illumina NovaSeq platform. The median (of medians) coverage achieved was 161× (min 43×, max 218×) (see Supplementary Data 3).

Trimming was performed with Skewer v0.1.126. Reads with a mean quality value > 10 prior to trimming and a minimum read length of 35 following trimming were kept. All others were discarded. Trimmed reads were aligned to the full human reference genome hg19 with the Burrows–Wheeler Aligner tool (bwa-mem, v0.7.15). PCR duplicates were marked using Picard tools (v2.8.1). Mutations were jointly called for all samples together using Platypus v0.8.1[63]. The extent of selection was determined by identifying SNVs exhibiting a 10× enrichment in VAF in the treated lines (GEF1–GEF3, TRM4–TRM6) over the POT line. This analysis yielded a cluster of four SNVs exhibiting enrichment corroborating that predicted by the barcode enrichment analysis. See Supplementary Data 4 for SNV calls filtered for a minimum coverage of 10 reads in all samples and a location within the target regions of the exome capture panel.

Heterozygous single nucleotide polymorphisms (SNPs) in the exome sequencing of the cell lines were identified using allelecount v3.0.1 (www.github. com/cancerit/alleleCount). Here we counted bases at SNP locations that have a global minor allele frequency between 0.1 and 0.2 (min. genomic position 100,000 bp) in dbSNP build 132[64] and overlap with the target regions of the exome panel for all autosomes. We calculate B-allele frequency (BAF) by dividing the highest base pair count by the total coverage at the SNP loci. These values were randomly subtracted from 1 to simulate the random assignment of the A and B allele. Log R ratio (LRR) was calculated as the log base 2 of the coverage of each SNP loci normalised by subtracting the global median LRR value.

To identify segments of copy number alterations (CNAs) we smoothed and segmented the LRRs of each sample using DNAcopy (Seshan and Olshen, 2016, 'DNAcopy: DNA copy number data analysis'. R package version 1.48.0). In order to calculate the mean heterozygous major allele frequency in each segment we required a test for distinguishing between segments with pure loss-of-heterozygosity (LOH) and segments containing heterozygous SNPs. We identified segments with heterozygosity by counting the numbers of SNPs in each segment with a major allele frequency <0.9. We then performed an exact binomial test in which the alternative hypothesis was that more than 5% of the segment contains heterozygous SNPs ($p < 0.05$). For those segments in which the null hypothesis was rejected, the median heterozygous major allele frequency value was used to represent the allelic (im)balance of the segment.

Using the ASCAT equations[65], we assumed each sample was pure (rho = 1) and solved the ploidy of each sample (psi) by calculating the distance of the continuous major and minor copy number values of all segments from their nearest integer states across a range of psi values that are realistic for tumour ploidy (1.5–5.5). The psi value that produced the smallest distance from integers in all segments was

taken as the ploidy solution. This was ~3 for all cell lines as the cell line HCC827 is known to be triploid[40]. See Supplementary Data 1 for copy number calls.

We additionally calculated GC content normalised depth ratios between each treated cell line and the parental population (POT) using Sequenza[66]. To calculate segments of differential copy number status, we subset the loci by their global minor allele frequency in dbSNP build 132 and segmented the depth ratios using DNAcopy. Segments in the depth ratio analysis are considered gains if the depth ratio is >1.2 and losses if <0.8.

**Digital droplet PCR**. Genomic DNA isolation for ddPCR was performed using DNeasy Blood and Tissue DNA extraction kit (Qiagen) according to manufacturer's recommendations. Quantification of gDNA was carried out using Qubit (Life Technologies). Digital droplet PCR (ddPCR) was performed on a QX200 ddPCR machine (Bio-Rad). Copy number assay was performed using 3 ng gDNA as a template and commercially available probes for *MET* (dHSACP2500321, FAM, Bio-Rad), *CDKN2A* (dHSACP1000581, FAM, Bio-Rad) and *NSUN3* (dHSACP2506682, HEX, Bio-Rad) as a reference gene. PCR reactions were performed using 3 ng of DNA, 10 μl of 2xSupermix in a total volume of 20 μl. Automated droplet generator (Bio-Rad) was used to generate ~20,000 droplets for partition of PCR reactions. Negative controls with no DNA and positive control DNA extracted from a cell line with previously reported CN were included. QuantaSoft v1.3.2.0 software was used for *MET* and *CDKN2A* CN analysis. Copy number status of *NSUN3* was assumed to be 3 (triploid) and this was confirmed by copy number analysis in exome sequencing data.

**High-throughput drug screening**. Cells from DMSO7 F4, TRM4 D6 and E9 were tyripsined and counted. 120–600 cells per well were seeded in 384-well plates (Corning). Cells were grown in a 37 °C and 5% $CO_2$ incubator overnight. A panel of 485 agents (Supplementary Data 5) was prepared in three different concentrations (20, 200 and 800 nM) and dispensed per well using Echo 555 liquid handler (Labcyte Inc.). After 5 days of treatment with agents, cells were incubated with 10% CellTiter-Blue cell viability reagent (Promega) for 4 h in a 37 °C and 5% $CO_2$ cell culture incubator. Finally, EnVision (PerkinElmer) plate reader was used to obtain readings.

Hit identification was performed separately for each of the three drug concentrations and three replicates of each line. First, raw fluorescence intensities were converted into an estimated percentage of inhibition (PCI), using the following formula:

$$\mathrm{PCI} = \frac{100 \times (c_{pos} - I)}{c_{pos} - c_{neg}}$$

where $I$ is the raw fluorescence intensity, $c_{pos}$ is the positive control of the plate, and $c_{neg}$ is the negative control of the plate. The plate-specific positive control was defined as the average fluorescence of 14 wells seeded with cells but no drug. The plate-specific negative control was defined as the average fluorescence of 14 empty wells.

Collaterally sensitive second line therapies were identified as those exhibiting a large increase in mean PCI, with respect to the DMSO lines, across multiple replicates and/or concentrations. Compounds were ranked, for a given replicate and concentration, according to their mean PCI change. Those with PCI change <5% at the given concentration were excluded and the top six of those remaining were considered for hit identification. Potential hits were identified as compounds appearing in the top six of more than one concentration or replicate.

**Dose response curves**. Resistant cell lines and single cell clones were trypsinised and counted. Between 300 and 10,000 cells per well were seeded in 96-well standard plates (Corning). Following overnight incubation in a 37 °C and 5% $CO_2$ cell culture incubator, average 10-fold changing dose of 10 concentrations from each inhibitor were used. 3 days post inhibitor treatment for all of the drugs validated, with an exception of 10 days for trametinib and 7 days for capmatinib. 10% CellTiter-Blue cell viability reagent (Promega) was applied. After overnight of incubation with 10% CellTiter-Blue in a 37 °C and 5% $CO_2$ cell culture incubator, readings were obtained using EnVision (PerkinElmer) plate reader.

To derive dose–response curves, normalised percentage growth was derived from OD readings by normalisation to six positive control (drug-free growth) and six negative control (empty) wells. A two parameter (ec50, hill coefficient) log-logistic dose response curve was then fitted to the data via non-linear least-squares regression.

**Luminex phosphoprotein assay**. POT, DMSO7, GEF1 and TRM4 cell lines were tyripsinied and counted. Following seeding of 300,000 cells per well in six-well plates and incubation in a 37 °C and 5% $CO_2$ cell culture incubator overnight, three biological replicates of each cell lines were treated with DMSO, 40 nM of gefitinib and 100 nM of trametinib for 1 h. After the incubation under those conditions, cells were tyripsinised and centrifuged at 1500 rpm to generate cell pellets. Cell pellets were lysed using MDS Tris Lysis Buffer (Meso Scale Diagnostics) containing phosphatase inhibitor I (Sigma-Aldrich), phosphatase inhibitor II (Sigma-Aldrich), protease inhibitor (Cell Signalling Technology). Protein content of lysed samples

was quantified using BCA assay (Sigma-Aldrich). MILLIPLEX MAP Akt/mTOR phosphoprotein kit, MILLIPLEX MAPK/SAPK signalling kit, MILLIPLEX MAP RTK phosphoprotein kit (48-611MAG, 48-660MAG, HPRTKMAG-01K respectively, MerckMillipore) were combined with the following singleplex magnetic bead sets to produce three multiplex Luminex assays; Total HSP27, GAPDH (46-702MAG, 46-710MAG, 46-623MAG, 46-641MAG, 46-608MAG, 46-667Mag, MerckMilipore). Bio-Plex Pro phosphor-PDGFRb and Akt (Thr308) (171-V50018M, 171-V50002, Bio-Rad) were combined into a triplex assay. Manufacturer's recommendations were followed. Phosphoprotein levels were measured on the Luminex 200 system utilising xPOTENT c3.1 software.

**Floating barcodes harvesting**. To track evolution through time, we leveraged the large volume of media (560 ml) that must be changed each week to maintain the HYPERflask culture system. HCC827 is an adherent cell line, with cells that detach from the plate surface upon death. By spinning the spent media in a centrifuge at 1,200 rpm for 10 min, we collected pellets consisting of cells that had died within the week. We extracted barcodes from these intermediate time points for each of the gefitinib exposed lines (weekly for 4 weeks) and for each of the trametinib-resistant exposed lines (weekly for 9 weeks). These barcodes permitted us to track the evolution of each cell lineage, under each drug exposure, without the need for re-plating, and with a temporal resolution that is unparalleled. Apoptotic barcoded cells were extracted using DNeasy Blood and Tissue DNA extraction kit (Qiagen).

**Single cells RNA profiling**. Single cells were prepared from POT, GEF1 and TRM4 cells. After centrifugation, single cells were washed with PBS and were re-suspended with a buffer ($Ca^{++}/Mg^{++}$ free PBS + 0.04% BSA) at 1000 cells/μl.

Viability was confirmed to be >90% in all samples using acridine orange/propidium iodide dye with LUNA-FL Dual Fluorescence Cell Counter (Logos Biosystems, L20001). Single cell suspensions were loaded on a Chromium Single Cell 3′ Chip (10X Genomics) and were run in the Chromium Controller to generate single-cell gel bead-in-emulsions using the 10X genomics 3′ Chromium v2.0 platform as per manufacturer's instructions. Single-cell RNA-seq libraries were prepared according to the manufacturer's protocol and the library quality was confirmed with a Bioanalyzer High-Sensitivity DNA Kit (Agilent, 5067-4627) and a Qubit dsDNA HS Assay Kit (ThermoFisher, Q32851). Samples were pooled up to three and sequenced on an Illumina HiSeq 4000 according to standard 10X Genomics protocol.

CellRanger (v2.1.1) was run on the raw data using GRCh38 annotation (v1.2.0). Output from cellRanger was loaded into the statistical computing environment R v3 (www.r-project.org) through the function load_cellranger_matrix_h5 from package *cellranger* (v1.1.0; genome = "GRCh38"). Datasets were merged according to gene names. Before normalisation, a series of filtering steps was performed. Only those cells showing at least 1500 detected genes and 5000 UMIs were considered for further analyses[67]. Reads mapping on mitochondrial genes were excluded. After that, data were imported in *Seurat* (v2.3.4)[68] and scaled (*NormalizeData* function using normalization.method = "LogNormalize", scale.factor = 10,000, followed by the *ScaleData* function). A further filtering step was performed based on the cumulative level of expression (the sum of the Seurat-scaled values) of three housekeeping genes (*GAPDH*, *RPL26* and *RPL36*)[69]. Manual inspection of these values versus the number of UMIs per cell (or the number of genes with non-zero expression per cell) revealed no significant correlation between the two. Nevertheless, a number of cells showed extremely low expression of these genes, so those in the bottom 1% were excluded from further analyses. At last, genes expressed in <20 cells were also excluded. Linear normalisation and scaling were performed again on the filtered, raw data. Variable genes were identified using the *FindVariableGenes* function of *Seurat* (mean.function = ExpMean, dispersion.function = LogVMR, x.low.cutoff = 0.01, x.high.cutoff = 6, y.cutoff = 0.01, num.bin = 100). Principal component analysis (PCA) was run on variable genes as input and, based on *p*-values estimated by the *JackStraw* function, the top 44 components were kept. These components were used as input for further dimensionality reduction (using *t*-Distributed Stochastic Neighbour Embedding; t-SNE) through the *RunTSNE* function (perplexity = 50, do.fast = TRUE, seed.use = 44). Clusters were then identified using *FindClusters* (resolution = 0.6).

**Single clones isolation**. Single cell isolation from DMSO7, GEF1 and TRM4 cell lines was performed using CellenONE™ (Scienion, Lyon). Harvested cells were diluted in PBS to generate cell suspension. Later, the number of cells in the suspension were controlled optically in the piezo dispense capillary (PDC) to manage the presence of truly single cell for each dispensing. Finally, each of the single cells was dispensed into a specific position in the 96-well flat bottom microplate already filled with 100 μl of full growth media. Ultimately, single cells grown in 96-well plate were harvested and they were transferred into a cell culture flask for their expansion. Single-cell-derived cell lines were named according to their position in the 96-well plate that they have originated.

**Reporting summary**. Further information on research design is available in the Nature Research Reporting Summary linked to this article.

## Data availability

Exome sequencing data has been deposited at the European Genome-phenome Archive (EGA), which is hosted by the EBI and the CRG, under accession number EGAS00001003200. Further information about EGA can be found on https://ega-archive.org. Single cell sequencing data is deposited in both raw and processed form in ArrayExpress under the Accession E-MTAB-8809. FASTQ files containing the sequencing of barcodes are deposited in ArrayExpress under the Accession E-MTAB-8841. ArrayExpress is also hosted by EMBL-EBI and the data can be found at www.ebi.ac.uk/arrayexpress.

## Code availability

Python code used to analyse the barcodes and to create plots in Fig. 4 are hosted at www.github.com/sottorivalab/ExpEvolutionSteering.

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

## Acknowledgements

A.S. is supported by the Wellcome Trust (202778/B/16/Z) and Cancer Research UK (A22909). We acknowledge funding from the National Institute of Health (NCI U54 CA217376) to A.S. and C.C.M. This work was also supported by Wellcome Trust award to the Centre for Evolution and Cancer (105104/Z/14/Z). U.B. is supported by The NIHR (RP-2016-07-28), CRUK (A25128, A22897) and by the Experimental Cancer Medicine Centre grant and the NIHR Biomedical Research Centre grant to The Institute of Cancer Research and The Royal Marsden NHS Foundation trust. C.C.M. was supported in part by NIH grants U54 CA217376, P01 CA91955, R01 CA170595, R01 CA185138 and R01 CA140657 as well as CDMRP Breast Cancer Research Programme Award BC132057 and an Arizona Investigator Grant ADHS18-198847. L.M. was supported by Cancer Research UK (A23110). NV is supported by Cancer Research UK (A18052 and A26815), the National Institute for Health Research (NIHR) Biomedical Research Centre (BRC) at The Royal Marsden NHS Foundation Trust and The Institute of Cancer Research (grant numbers A62, A100, A101, A159) and the European Union FP7 (CIG 334261). We thank Nik Matthews and the Tumour Profiling Unit at the ICR for their support with Next-Generation Sequencing. We also thank Michael Hubank, Eleni Koutroumanidou and Debbie Hughes for technical support.

## Author contributions

A.A. conducted the experiments. D.N. performed data analysis and mathematical modelling. A.A., D.N., U.B. and A.So. designed the experiments. J.F.M. assisted and contributed to experimental procedures. G.D.C., I.B., N.T., G.C., B.W. contributed to data analysis and modelling. A.A., D.N., G.D.C., U.B. and A.So. interpreted the results. I.S. generated bulk sequencing data. S.P.H. generated single-cell sequencing data. M.S. and R.B. contributed to designing and performing drug screening experiments. A.St. generated phosphoproteomic data. G.V. contributed to experimental procedures. U.B. and A.So. conceived and supervised the study. C.C.M., L.M. and N.V. contributed to study supervision. A.So. and U.B. wrote the manuscript. All authors contributed to manuscript preparation and editing.

## Competing interests

U.B. has received funding for investigator-initiated clinical trials for a phase I study of vistusertib and paclitaxel from Astra Zeneca. U.B. has received funding for investigator initiated clinical trials for a phase I study of a dual MEK-RAF inhibitor RO5126766 from Chugai. NV received honoraria for lectures from Merck Serono, Pfizer, Bayer and Eli-Lilly. Neither company has funded or had input into research presented in this manuscript. All other authors declare no competing interests.
