## [Peer Review File · Nature Communications]

Editorial Note: This manuscript has been previously reviewed at another journal that is not operating a transparent peer review scheme. This document only contains reviewer comments and rebuttal letters for versions considered at Nature Communications. Mentions of the previous journal have been redacted as indicated.

Reviewers' comments:

Reviewer #1 (Remarks to the Author): This reviewer was Reviewer #1 for [redacted]; evolution of resistance and trade-offs

The authors have failed to convince me that the comments made on the [redacted] version of this manuscript are not relevant. I fail to see the fundamental difference between this manuscript and the earlier barcoding manuscripts. There are no new biological insights gained from this study.

Reviewer #4 (Remarks to the Author): Recruited to replace Reviewer #3; modelling and clonal evolution

Acar and colleagues describe in this manuscript an experimental set up to study growth dynamics of multi-clonal cancer cells under treatment. The hyperflask setting is certainly intriguing as it allows to grow unprecedentedly large cell populations and, thus, to test shifts of clone fractions under treatment in reproducible experiments.

The authors provide here an interesting proof of principle of this approach using a NSCLC cell line with a previously reported subclone harboring a MET amplification, which leads to resistance to the EGFR inhibitor Gefitinib. I'm not sure whether the CDKN2A deletion-subclone was also previously reported, but the data shown in this paper strongly support it as pre-existing (although it is not conclusive).

Although I have the feeling that more could have been done to prove the power of this approach (the results supporting the value of this approach to discover effective sequential treatments are limited, e.g. validation experiments of the drug screening would have been nice) I understand this manuscript has already been reviewed and therefore I do not intend to ask for substantial new experiments or analyses. Overall, I have no strong objections against its publication in Nature Communications.

I'd like to offer either way a few suggestions and minor comments that I'd ask the authors to address:

1) As mentioned above, the main limitation of this study is that it remains a proof of principle with no new results, except for the approach itself. Given that, it would be helpful to provide/discuss more extensively guidelines on when and how this approach is meaningful and should be used. For example, what happen if a cell line is clonal, can this approach still be used to study the emergence of new mechanisms of resistance in a faster way, given the large sample size? Is it a particularly suited system to be combined with high-throughput screening like the one performed in Figure 6? Do the author envision it could be expanded to grow immortalized cells from primary tumors (which could display higher heterogeneity than cell lines) or the condition are too difficult to grow such models?

2) Figure 1, 2, and 3 are almost exclusively representative diagrams / toy images of the approaches here taken, we only get to actual data in Figure 4. This seems a bit odd. I suggest condensing this information into 1 figure and move most of the panels to the supplemental material (e.g. Figure 2). It is my opinion that 4 figures in total are here more than enough.

3) The authors talk about mathematical modeling in the abstract and on page 8, the latter in reference to the estimated growth rates. The only mathematical model that I could find described in this manuscript is the one used to estimate the time of emergence of resistant mutations in Figure 1D. Growth rates are estimated from measured barcode frequencies using a simple log fold-change expression (in the methods the authors indeed talk about "bioinformatics analysis"). Unless I missed something (my apologies if so), I would remove the term 'mathematical modeling' from the abstract and the text (except maybe for the data shown in Figure 1D).

4) "Moreover, cell plasticity and drug tolerance, instead of Darwinian adaptation, often occurs in current model systems, leading to resistance that is non-heritable, potentially reversible, and that does not represent what happens in the clinic. Non-heritable drug resistance can arise through epithelial-mesenchymal transition³⁴ or upregulation of drug-efflux pumps³⁵. Although these are very important cellular mechanisms of resistance, they do not pertain to clonal evolution, which drives persistent resistance in human cancers over long timescales"

This is clearly an overstatement and a misleading/self-contradicting sentence. Saying that cell plasticity does not represent what happens in the clinic and then mention EMT as an example is just wrong. Cell plasticity is relevant in the clinic, as is EMT, it is heritable (in a cancer pertaining sense, i.e. transmissible through cell generations), these mechanisms do pertain to clonal evolution as demonstrated by the authors themselves in Figure 6B and the argument on 'persistent resistance' although reasonable is not proven. The authors should just remove this paragraph (which is unnecessary) or substantially correct it.

5) It is surprising to see in Figure 6F that DMSO and POT do not respond to Gefinitib, given this should correspond to the initial population that is sensitive to this drug. Is this dose related? Can the authors comment on that?

Given these resistant clones are pre-existing, one is left to wonder whether a combination of MET and EGFR inhibitors (at different concentrations) would have been effective from the beginning or whether the sequential treatment would have still been better. This could be a simple and interesting experiment to do and could reinforce the relevance of the approach.

5bis) Related to this: A deep targeted sequencing of the cell line here used would have led to the discovery of MET and CDKN2A alterations? (both MET and CDKN2A are in many targeted panels)

6) Minor typos:

- Page 8: "As this group comprises of barcodes" should be "As this group comprises barcodes" or "As this group is composed of barcodes"

- Page 11: "CDK4/5 inhibitor" should be "CDK4/6 inhibitor"

- Fig 6G: legend is missing the label, what do these colors/numbers represent?

Reviewer #5 (Remarks to the Author): Recruited to replace Reviewer #2; lung cancer and EGFR signalling

The authors have submitted a revised manuscript and response letter. This reviewer was asked to assess the comments of reviewer #2 (original). The authors have done a good job in addressing the comments. The main issue this reviewer sees with the revised manuscript is the rather incremental advance of the study compared to the prior literature - as commented upon by both reviewer #1 and #3.

Reviewers' comments:

We thank all reviewers again for their assessment of our manuscript and constructive criticisms. Below is the point by point response to reviewer #4 comments.

Reviewer #4 (Remarks to the Author): Recruited to replace Reviewer #3; modelling and clonal evolution

Acar and colleagues describe in this manuscript an experimental set up to study growth dynamics of multi-clonal cancer cells under treatment. The hyperflask setting is certainly intriguing as it allows to grow unprecedentedly large cell populations and, thus, to test shifts of clone fractions under treatment in reproducible experiments. The authors provide here an interesting proof of principle of this approach using a NSCLC cell line with a previously reported subclone harboring a MET amplification, which leads to resistance to the EGFR inhibitor Gefitinib. I'm not sure whether the CDKN2A deletion-subclone was also previously reported, but the data shown in this paper strongly support it as pre-existing (although it is not conclusive).

We thank the reviewer for appreciating the novelty of the approach presented in this study. We do confirm that to the best of our knowledge, CDKN2A loss has never been associated to trametinib resistance. We highlight this more clearly in our revised version of the manuscript (page 6, text in red).

Although I have the feeling that more could have been done to prove the power of this approach (the results supporting the value of this approach to discover effective sequential treatments are limited, e.g. validation experiments of the drug screening would have been nice) I understand this manuscript has already been reviewed and therefore I do not intend to ask for substantial new experiments or analyses. Overall, I have no strong objections against its publication in Nature Communications. I'd like to offer either way a few suggestions and minor comments that I'd ask the authors to address:

1) As mentioned above, the main limitation of this study is that it remains a proof of principle with no new results, except for the approach itself. Given that, it would be helpful to provide/discuss more extensively guidelines on when and how this approach is meaningful and should be used. For example, what happens if a cell line is clonal, can this approach still be used to study the emergence of new mechanisms of resistance in a faster way, given the large sample size? Is it a particularly suited system to be combined with high-throughput screening like the one performed in Figure 6? Do the authors envision it could be expanded to grow immortalized cells from primary tumors (which could display higher heterogeneity than cell lines) or the conditions are too difficult to grow such models?

This is a very interesting point. Indeed, in a clonal cell line with no pre-existing resistance one would expect that all resistance dynamics are driven by drug tolerance followed by de novo resistance. These dynamics are often described by bet-hedging phenomenon (Nichol et al. 2016 – Genetics). This could be definitely studied with the presented platform, although one would expect different barcodes in different replicas, making the evolution highly stochastic. On the other hand, the floating barcodes allow to determine the waiting time of a de novo mutant with great precision, and hence measure the temporal dynamics in the context of bet-hedging, which are key to understand mutation rates and dynamics of resistance. We discuss this aspect further in the revised version of the manuscript (page 13, paragraph in red). The point of whether this approach can be extended to patient-derived lines is a great point and we discuss this potential application in the revised version of the manuscript (page 13, paragraph in red).

2) Figure 1, 2, and 3 are almost exclusively representative diagrams / toy images of the approaches here taken, we only get to actual data in Figure 4. This seems a bit odd. I suggest condensing this information into 1 figure and move most of the panels to the supplemental material (e.g. Figure 2). It is my opinion that 4 figures in total are here more than enough.

We agree with this reviewer. We have combined Figure 1 and 2 into a single figure. Given the complexity of the experimental assay and for clarity, we have left old Figure 3 on its own however.

3) The authors talk about mathematical modeling in the abstract and on page 8, the latter in reference to the estimated growth rates. The only mathematical model that I could find described

in this manuscript is the one used to estimate the time of emergence of resistant mutations in Figure 1D. Growth rates are estimated from measured barcode frequencies using a simple log fold-change expression (in the methods the authors indeed talk about "bioinformatics analysis"). Unless I missed something (my apologies if so), I would remove the term 'mathematical modeling' from the abstract and the text (except maybe for the data shown in Figure 1D).

We do agree that we do not sufficiently expand on the importance of mathematical modelling in the manuscript. However, we maintain it is key to do simulations with stochastic branching processes to perform experimental design, hence our introductory work in Figure 1, Material and Methods (first section) and Supplementary Note. Following from that it's the case that we simplified the maths to simply calculate growth rates, since the time-course barcodes give enough information to fit a simple model. In light of this comment and others, we have extended the part on mathematical modelling of the experimental design and barcodes (see Material and Methods, page 16, text in red and new Figure S2). Given the mathematical modelling embedded at the core of the design of the whole experiment (see Figure 1), we kept the term "mathematical modelling" in the abstract.

4) "Moreover, cell plasticity and drug tolerance, instead of Darwinian adaptation, often occurs in current model systems, leading to resistance that is non-heritable, potentially reversible, and that does not represent what happens in the clinic. Non-heritable drug resistance can arise through epithelial-mesenchymal transition³⁴ or upregulation of drug-efflux pumps³⁵. Although these are very important cellular mechanisms of resistance, they do not pertain to clonal evolution, which drives persistent resistance in human cancers over long timescales"

This is clearly an overstatement and a misleading/self-contradicting sentence. Saying that cell plasticity does not represent what happens in the clinic and then mention EMT as an example is just wrong. Cell plasticity is relevant in the clinic, as is EMT, it is heritable (in a cancer pertaining sense, i.e. transmissible through cell generations), these mechanisms do pertain to clonal evolution as demonstrated by the authors themselves in Figure 6B and the argument on 'persistent resistance' although reasonable is not proven. The authors should just remove this paragraph (which is unnecessary) or substantially correct it.

Indeed, we fully agree with this reviewer and we apologise for the confusion. We have now changed the introduction and the conclusions to highlight the importance of cell plasticity and heritable non-genetic changes such as EMT (page 2, text in red).

5) It is surprising to see in Figure 6F that DMSO and POT do not respond to Gefitinib, given this should correspond to the initial population that is sensitive to this drug. Is this dose related? Can the authors comment on that?

Given these resistant clones are pre-existing, one is left to wonder whether a combination of MET and EGFR inhibitors (at different concentrations) would have been effective from the beginning or whether the sequential treatment would have still been better. This could be a simple and interesting experiment to do and could reinforce the relevance of the approach.

5bis) Related to this: A deep targeted sequencing of the cell line here used would have led to the discovery of MET and CDKN2A alterations? (both MET and CDKN2A are in many targeted panels)

This is a good point that concerned us in the original manuscript. In Supplementary Figure 4 we performed digital droplet PCR, which is ~10 times more sensitive than targeted sequencing, to assess whether MET amplification and CDKN2A loss was detectable in the original POT population. The issue is that contrary to point mutations, copy number alterations are read as an average of the copies per cell in a bulk sample, hence ddPCR nor sequencing can pick up low frequency copy number changes in those samples. However, we do note that in the scRNA-seq experiment, a small subset of single cells with MET amplification and CDKN2A loss are found in the original POT (see Supplementary Figure 12).

It is indeed surprising that combination of gefitinib and captamatinib shows decreased sensitivity in POT with respect to gefitinib alone. We speculate that given the consistent sensitive profiles for gefitinib in this line, this is possibly due to drug antagonism. We report this in the revised manuscript.

6) Minor typos:

- Page 8: "As this group comprises of barcodes" should be "As this group comprises barcodes" or "As this group is composed of barcodes"

Sorry, corrected.

- Page 11: "CDK4/5 inhibitor" should be "CDK4/6 inhibitor"

Sorry, corrected.

- Fig 6G: legend is missing the label, what do these colors/numbers represent?

Sorry, corrected.